# Bioelectrical Impedance Analysis for Preoperative Volemia Assessment in Living Donor Hepatectomy

**DOI:** 10.3390/jpm12111755

**Published:** 2022-10-22

**Authors:** Suk-Won Suh

**Affiliations:** Department of Surgery, Chung-Ang University College of Medicine, Chung-Ang University Hospital, 224-1, Heuk Seok-Dong, Dongjak-Ku, Seoul 156-755, Korea; bumboy1@cau.ac.kr; Tel.: +82-2-6299-3184; Fax: +82-2-824-7869

**Keywords:** living donor hepatectomy, bioelectrical impedance analysis, preoperative volemia assessment, surgical field, donor safety

## Abstract

Donor safety remains an important concern. We introduced preoperative bioelectrical impedance analysis (BIA) in living donor hepatectomy, as it is a practical method for volemia assessment with the advantages of noninvasiveness, rapid processing, easy handling, and it is relatively inexpensive. We analyzed 51 living donors who underwent right hemihepatectomy between July 2015 and May 2022. The ratio of extracellular water:total body water (ECW/TBW; an index of volemic status) was measured. ECT/TBW < 0.378 was correlated to central venous pressure (CVP) < 5 mm Hg in a previous study and we used this value for personalized preoperative management. In the BIA group (n = 21), donors with ECW/TBW ≥ 0.378 (n = 12) required whole-day nothing by mouth (NPO), whereas those with ECW/TBW < 0.378 (n = 9) required midnight NPO, similar to the control group (n = 30). In comparison with the control group, the BIA group had a significantly lower central venous pressure (*p* < 0.001) from the start of surgery to the end of surgery, leading to a better surgical field grade (*p* = 0.045) and decreased operative duration (240.5 ± 45.6 vs. 276.5 ± 54.0 min, *p* = 0.016). A cleaner surgical field (surgical field grade 1) was significantly associated with decreased operative duration (*p* = 0.001) and estimated blood loss (*p* < 0.001). Preoperative BIA was the only significant predictor of a cleaner surgical field (odds ratio, 6.914; 95% confidence interval, 1.6985–28.191, *p* = 0.007). In conclusion, preoperative volemia assessment using BIA can improve operative outcomes by creating a favorable surgical environment in living donor hepatectomy.

## 1. Introduction

As the demand for liver transplantation (LT) is increasing, living donor LT (LDLT) has emerged as a valuable alternative option to deceased donor LT (DDLT) [1]. Currently, LDLT is performed at a greater frequency than that of DDLT due to the limited availability of deceased donor organs, especially in Asian populations [2]. LDLT is a preferred treatment for selected hepatocellular carcinoma (HCC) patients because it can treat both the tumor and underlying liver diseases including hepatitis B viral infection, which increases the risk of tumor recurrence after hepatic resection [3,4]. In LDLT, donor safety remains a major concern because living donors are typically healthy adults who do not derive any medical benefit from the procedure. It is important to make consistent efforts to minimize the potential risks to the living donors to justify such an operation.

Excessive blood loss during hepatic resection has been reported to be significantly associated with postoperative morbidity and mortality [5]. Bleeding from the hepatic vein, which drains directly into the inferior vena cava (IVC), can be severe and cannot be controlled by portal triad clamping [6]. A low central venous pressure (CVP) during hepatic resection has been a widely used approach to reduce surgical blood loss and improve clinical outcomes [7]. Hepatic blood congestion, induced by elevated CVP, leads to an incremental increase in the transmural pressure and the distension of the hepatic vein, which is consequently torn easily, promoting blood loss at the time of parenchymal transection [8]. Preoperative fluid restriction is a widely used method for lowering the CVP; however, it is sometimes not sufficient, which makes hepatic resection difficult [9]. 

Bioelectrical impedance analysis (BIA) is a practical method for volemia assessment, which has the advantages of noninvasiveness, rapid processing, easy handling, and it is relatively inexpensive [10,11]. This method evaluates the human body composition, and the ratio of extracellular water to total body water (ECW/TBW) can be calculated as an index of volemic status [12]. A previous study showed that ECW/TBW and CVP were significantly correlated, which can be used preoperatively to significantly reduce the intraoperative bleeding during hepatic resection [13]. However, no previous study has focused on the clinical application of BIA for the preoperative volemia assessment of living donors. Here, we introduced BIA in living donor hepatectomy.

This study aimed to investigate the usefulness of preoperative BIA for maintaining a low CVP during hepatic resection, to improve operative outcomes in living donor hepatectomy.

## 2. Patients and Methods

### 2.1. Patients

The study population consisted of 51 consecutive living donors who underwent right hemihepatectomy for LDLT between March 2015 and May 2022 at our hospital. The donors were selected according to a standardized protocol described previously in [14]. The first 30 donors were enrolled in the control group, and the next 21 donors who underwent BIA for preoperative volemia assessment were enrolled in the BIA group. In the BIA group, the time of nothing by mouth (NPO) was decided based on the value of ECW/TBW, measured 1 day before surgery after admission; ECW/TBW < 0.378 was correlated to CVP < 5 mm Hg in a previous study [13]. Donors with ECW/TBW ≥ 0.378 (n = 12) required whole-day NPO, whereas those with ECW/TBW < 0.378 (n = 9) required midnight NPO, similar to the control group (Figure 1). Demographics and intraoperative and postoperative outcomes were compared between the groups.

### 2.2. Data Collection

Age, sex, presence of diabetes mellitus or hypertension, body mass index (BMI), type of graft, degree of hepatic steatosis, postoperative remnant liver volume, baseline liver function [total bilirubin (TB), international normalized ratio (INR), and albumin], and baseline renal function (creatinine) at admission were collected for all living donors. Operative details, such as operative duration, fluids administered including crystalloid and colloid, urine output, estimated blood loss, requirement for blood transfusion, inotropic use, and changes in CVP during hepatic resection were collected. The surgical field was assessed using a previously reported 4-point scale [15]: grade 1 = very lax IVC and minimal bleeding in the resection plane; grade 2 = lax IVC and a little bleeding in the resection plane; grade 3 = tense IVC and appreciable bleeding in the resection plane; grade 4 = very tense IVC, profuse bleeding in the resection plane, and very difficult to operate. Perioperative laboratory results, including TB, INR, albumin, and creatinine were analyzed. In addition, all available intake records, such as oral and parenteral fluids and output data including urine, gastrointestinal losses, and drains from the operative day to postoperative day 7 were collected. Acute kidney injury (AKI) was defined in accordance with the 2012 Kidney Disease Improving Global Outcomes guidelines, which have higher predictability compared with other criteria for assessing prognosis [16]; the guidelines were as follows: increase in serum creatinine by ≥0.3 mg/dL within 48 h, increase in serum creatinine to ≥1.5 times baseline within 7 days before surgery, or urine volume < 0.5 mL/kg/h for 6 h. Postoperative liver insufficiency was defined as a peak postoperative TB level > 7 mg/dL and/or the presence of ascites > 500 mL/day based on a previous study [17].

### 2.3. BIA

Body composition was assessed using a segmental and multifrequency BIA device (InBody S10; Biospace, Seoul, Korea) 1 day before surgery, after admission in the BIA group. Eight electrodes were placed on the surface of the thumb, fingers of the hand, and ball of the foot and heel with the patients in the supine position. We calculated the ECW/TBW ratio to estimate the volemic status of the patients. Three repeated measurements of BIA were performed, and the median value was chosen in order to avoid an incorrect measurement.

### 2.4. Anesthetic and Surgical Technique

Anesthetic management was performed using a standard protocol in our hospital. General anesthesia was induced with intravenous 100 μg of fentanyl and 1.2 mg/kg of propofol, followed by intravenous 1 mg/kg of rocuronium to facilitate endotracheal tube placement. General anesthesia was maintained with sevoflurane (2 to 3 volume%), nitrous oxide (1.8 L/min), and O_2_ (1.2 L/min). The CVP was continuously monitored using right jugular vein catheterization. Electrocardiogram, pulse oximetry, end-tidal carbon dioxide, invasive radial arterial pressure, and urine output were also monitored. Fluid was not administered preoperatively and was restrictively infused after the start of anesthesia, maintaining CVP less than 5 mm Hg until the hepatic parenchymal transection was complete. Thereafter, the crystalloid fluid was rapidly infused at 10 to 12 mL/kg/h to replace the surgical blood loss or fluid deficit, including insensible loss during the operation, and a colloid solution, hydroxyethyl-starch (HES), was used considering the volume status in the operating room. We used vasopressor drugs when the MAP decreased below 60 mm Hg. Mostly, 5 mg bolus of ephedrine was administered, but if an elevated heart rate was present, 50 mcg bolus of phenylephrine were injected. Red blood cells were transfused if the hemoglobin concentration decreased to <7 g/dL during the operation and in the perioperative period.

The detailed surgical techniques for mobilization of the liver, dissection of the hilum and parenchyma, and graft retrieval have been described previously in [18]. After the completion of hemostasis, a closed suction drain was inserted into the right subphrenic space. The abdominal wall was closed layer by layer. After subepidermal suturing, a sterile strip was applied to the skin incision.

### 2.5. Statistical Analysis

Clinico-demographic characteristics of living donors, changes in CVP during the operation and intraoperative and perioperative outcomes of patients were compared using the Student’s *t*-test for distributed data, presented as means ± standard deviations and the χ2 test for descriptive data. The surgical field grades were compared using the Mann–Whitney U test. Univariate and multivariate analysis of risk factors for a cleaner surgical field was performed using an ordinary logistic regression model. Differences in operative outcomes according to the surgical field grade were compared using the Student’s *t*-test. *p* values < 0.05 were considered to indicate statistical significance. Statistical analysis was conducted using the statistical package for the social sciences (SPSS) version 19.0 (IBM Corp., Armonk, NY, USA).

## 3. Results

The clinico-demographic characteristics of the living donors are summarized in Table 1. There were no significant differences in age, sex, prevalence of diabetes mellitus or hypertension, body mass index, hepatic steatosis, remnant liver volume, baseline liver function including TB, albumin, and INR, and baseline renal function (creatinine) between the two groups. The expanded criteria donors (age ≥ 50, BMI ≥ 25, hepatic macrosteatosis ≥ 10%, and remnant liver volume < 30%) also showed no significant differences between the two groups.

BIA was selectively performed again on the operative day for donors with ECW/TBW ≥ 0.378 to analyze the changes in the value after whole-day NPO. Overall, 10 of 12 donors (83.3%) showed a decrease in the ECW/TBW value to below 0.378, whereas 2 of 12 donors (16.7%) still had an ECW/TBW value ≥ 0.378.

The CVP was significantly lower in the BIA group than in the control group from the start of surgery to the end of surgery (*p* < 0.001). In both groups, the CVP showed a decreasing trend after the start of surgery until the completion of hepatic parenchymal transection and it was increased after fluid challenge (Figure 2).

The mean operative duration (240.5 ± 45.6 vs. 276.5 ± 54.0 min, *p* = 0.016) and hepatectomy time were significantly shorter in the BIA group than in the control group (2167.5 ± 32.5 vs. 203.9 ± 42.6 min, *p* = 0.002). Estimated blood loss was decreased in the BIA group compared with the control group; however, the difference was not significant (325 ± 212 vs. 382 ± 210 mL, *p* = 0.349). There were no significant between-group differences in the intraoperative fluid administration of crystalloid and colloid, amount of urine output, and inotropic use. There was no requirement for blood transfusion in both groups. With respect to postoperative complications, there were no significant differences in the incidence rates of pleural effusion, AKI, and postoperative liver insufficiency between the two groups. There was no major postoperative complication in both groups. The duration of postoperative hospital stay was slightly decreased in the BIA group compared with the control group, without statistical significance (10.4 ± 2.2 vs. 10.8 ± 3.6 days, *p* = 0.679; Table 2).

The surgical field grade showed a significant difference between the groups. In the BIA group, grade 1 (71.4%) was the most common, followed by grade 2 (28.6%), whereas in the control group, grade 2 (60.0%) was the most common, followed by grade 1 (36.7%) and there was one case of grade 3 (3.6%) (*p* = 0.045; Figure 3).

Preoperative BIA was the only significant predictor for a cleaner surgical field (surgical field grade 1) in multivariate analysis (odds ratio, 6.914; 95% confidence interval, 1.699–28.191, *p* = 0.007; Table 3).

Differences in the operative duration, estimated blood loss, and postoperative hospital stay of donors according to the surgical field grade were analyzed. Operative duration (238.1 ± 55.3 vs. 286.2 ± 38.8 min, *p* = 0.001) and estimated blood loss (196 ± 57 vs. 528 ± 174 mL, *p* < 0.001) were significantly decreased among donors with surgical field grade 1 compared with those with surgical field grade ≥ 2; however, there was no significant difference in postoperative hospital stay between the two groups (10.7 ± 3.6 vs. 10.6 ± 2.5 days, *p* = 0.988; Figure 4).

## 4. Discussion

We investigated whether BIA is useful for preoperative volemia assessment in living donor hepatectomy. Donors with preoperative BIA had a significantly lower CVP during hepatic resection, which led to a better surgical field grade compared with those in the control group. Surgical field grade was significantly associated with decreased estimated blood loss and operative duration. Our results demonstrated that preoperative BIA can help improve the operative outcomes of living donors.

Maintaining a low CVP during hepatic resection significantly reduced intraoperative bleeding during hepatic resection in a previous study [8]. A low CVP is associated with low pressure in the hepatic vein and liver sinusoids, which reduces blood loss and allows easier control of bleeding in hepatic venous injury [19]. Although living donors with preoperative BIA had a significantly lower CVP with a better surgical field grade during hepatic resection compared with those in the control group, there was no significant difference in estimated blood loss between the two groups in this study. There might be several reasons for such discrepancies. First, some of the living donors maintained a low CVP only after midnight NPO; 11 of 30 donors (36.7%) in the control group had a CVP of <5 mm Hg in this study. Second, whole-day NPO was not sufficient for reducing the CVP; 2 of 12 donors (16.7%) in the BIA group with an ECW/TBW value of ≥0.378 at admission still had an elevated value on the operative day with a CVP of 6–7 mm Hg. Finally, the living donors had a normal healthy liver, which make hepatic parenchymal dissection and bleeding control easier than in other patients.

A study reported that BMI and hepatic steatosis were significant determinants of estimated blood loss in living donor hepatectomy [20]. Previously, obese donors were not considered for LDLT in many centers because some of them have moderate to severe hepatic steatosis, which may be associated with operative morbidity and mortality [21]. However, the use of expanded donor criteria has become more common owing to a considerable increase in waitlisted recipients and organ shortages in specific regions [2]. This study also included donors with BMI ≥ 25 (33.3%), the World Health Organization (WHO) definition of obesity for Asians [22]. We routinely performed magnetic resonance (MR) spectroscopy to screen for hepatic steatosis, and a weight reduction program was introduced to donors with ≥10% hepatic steatosis. They were given a scheduled protein-rich diet and performed exercises, and after several weeks of body weight reduction, follow-up MR spectroscopy was performed to evaluate the improvement in hepatic steatosis before surgery. Donors with a high BMI have increased ventilation pressure requirements, which can impair hepatic venous outflow and promote bleeding at the time of hepatic parenchymal transection [23]. Hepatic steatosis makes the liver tissue more frangible, which increases blood loss during hepatic resection [24]. Preoperative BIA for reducing the CVP during hepatic resection would be more important for these patients. In this study, preoperative BIA but not BMI and hepatic steatosis was the only significant predictor for surgical field grade ≥ 2.

BIA has many advantages for preoperative volemia assessment in living donor hepatectomy. First, it is a safe and noninvasive method that may replace CVP monitoring. It leads to a dry and cleaner surgical field, which is associated with a decrease in estimated blood loss; therefore, the placement of a central venous catheter is not required. Therefore, it can prevent catheter-related complications including arterial puncture, hematoma, pneumothorax, catheter-related infection, and thrombosis [25]. Second, it can quantify the volemic status of donors; thus, a personalized strategy for maintaining a low CVP suitable for hepatic resection may be possible. An exceedingly low CVP can increase the risk of the potential hypoperfusion of abdominal organs, causing AKI or air embolism of the lungs through hepatic venous injury during hepatic resection [26,27]. Third, it is easily and rapidly carried out with a portable instrument; thus, repetitive measurements to evaluate changes in the volemic status according to the duration of NPO are convenient.

There are some limitations in this study. This study was retrospective in nature, and the accuracy of the data analyzed relies on the completeness of medical records. Living donors in the control group received the operation before those in BIA group, so their experience of the operation might have influenced the operative outcomes. In addition, the study population was relatively small; thus, further large-scale prospective studies are warranted to confirm whether preoperative BIA can circumvent CVP monitoring without any increase in operative morbidity and mortality.

## 5. Conclusions

Living donors with preoperative BIA had a significantly lower CVP during hepatic resection, which led to a better surgical field grade and a decrease in the operative duration compared with those in the control group. A cleaner surgical field was significantly associated with decreased operative duration and estimated blood loss. Preoperative BIA was the only significant predictor a cleaner surgical field. Therefore, preoperative volemia assessment by BIA can help improve operative outcomes by creating a favorable surgical environment in living donor hepatectomy.

## Figures and Tables

**Figure 1 jpm-12-01755-f001:**
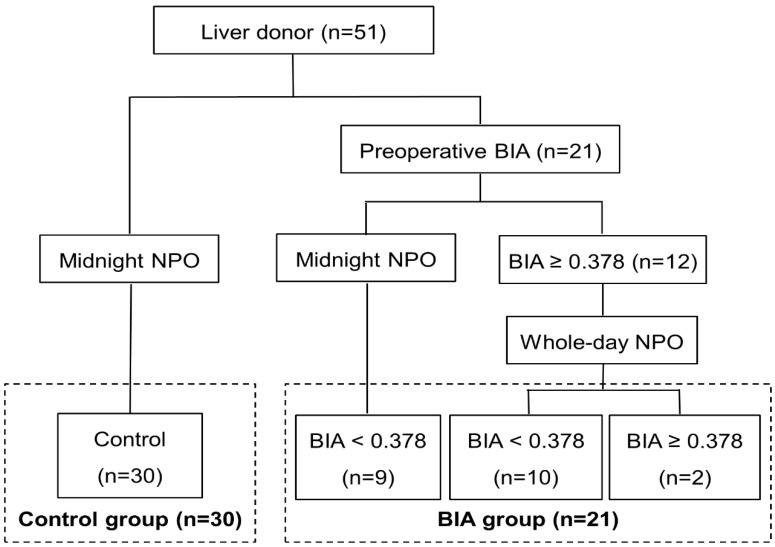
Flow diagram of the study. Among 51 living donors, the first 30 consecutive donors were enrolled in the control group, and the next 21 donors with preoperative BIA were enrolled in the BIA group. Donors with ECW/TBW ≥ 0.378 had whole-day NPO, whereas those with ECW/TBW < 0.378 had midnight NPO, similar to the control group. BIA was selectively performed again on the operative day for donors with ECW/TBW ≥ 0.378 to analyze the changes in the value after whole-day NPO. BIA, bioelectrical impedance analysis; NPO, nothing by mouth.

**Figure 2 jpm-12-01755-f002:**
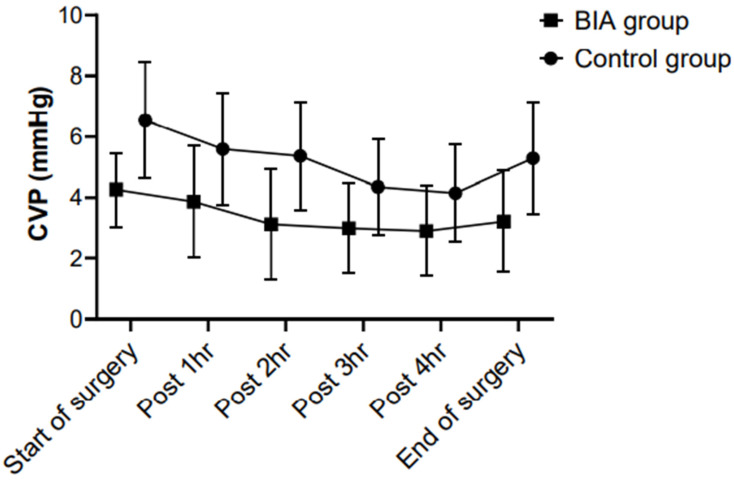
Between-group comparisons of the CVP during hepatic resection. The CVP was significantly lower in the BIA group than in the control group from the start of surgery to the end of surgery (*p* < 0.001). In both groups, the CVP showed a decreasing trend after the start of surgery until the completion of hepatic parenchymal transection and was increased after fluid challenge. CVP, central venous pressure; BIA, bioelectrical impedance analysis.

**Figure 3 jpm-12-01755-f003:**
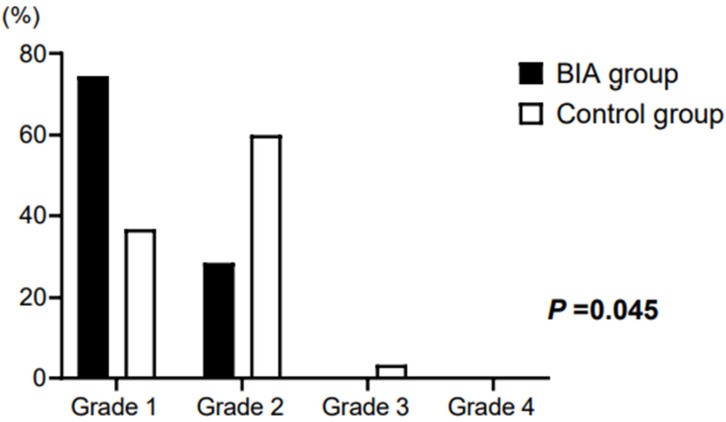
Surgical field grade of patients. In the BIA group, grade 1 (71.4%) was the most common, followed by grade 2 (28.6%). In the control group, grade 2 (60.0%) was the most common, followed by grade 1 (36.7%) and grade 3 (3.6%) (*p* = 0.045). BIA, bioelectrical impedance analysis.

**Figure 4 jpm-12-01755-f004:**
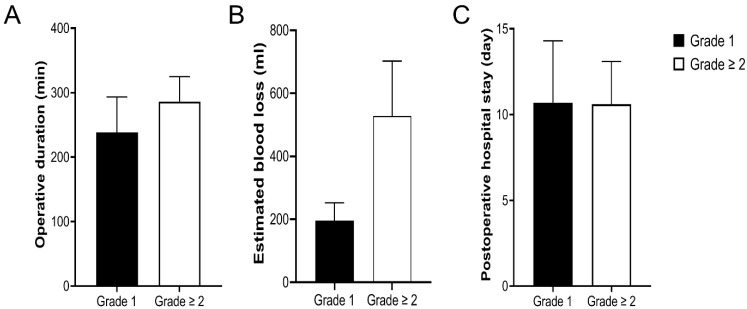
Differences in operative duration (**A**), estimated blood loss (**B**), and postoperative hospital stay (**C**) according to the surgical field grade. A significant decrease in operative duration (*p* = 0.001) and estimated blood loss (*p* < 0.001) was observed among donors with surgical field grade 1 compared with those with surgical field grade ≥ 2. However, there was no significant difference in postoperative hospital stay between the two groups (*p* = 0.988).

**Table 1 jpm-12-01755-t001:** Clinico-demographic characteristics of living donors.

	BIA Group (n = 21)	Control Group (n = 30)	*p*
Age, mean (years)	39.0 (±12.5)	36.9 (±10.2)	0.510
Age ≥ 50 (%)	3 (14.3%)	3 (10.0%)	0.640
Male (%)	11 (52.4%)	20 (66.7%)	0.304
Diabetes mellitus	0	2 (6.7%)	0.227
Hypertension	0	2 (6.7%)	0.227
BMI, kg/m^2^	23.4 (±2.7)	24.5 (±2.9)	0.187
BMI ≥ 25, (%)	6 (28.6%)	11 (36.7%)	0.546
Hepatic steatosis	1.4 (±2.3)	1.8 (±2.6)	0.574
Macrosteatosis ≥ 10%	0	1 (3.3%)	0.360
Remnant volume, mean (%)	35.0 (±3.3)	36.5 (±4.1)	0.163
Remnant volume < 30% (%)	1 (4.3%)	2 (7.1%)	0.673
Baseline liver function			
TB (mg/dL)	0.6 (±0.2)	0.6 (±0.2)	0.879
Albumin (g/dL)	4.6 (±0.2)	4.5 (±0.4)	0.476
INR	1.03 (±0.06)	1.04 (±0.06)	0.464
Baseline renal function			
Creatinine (mg/dL)	0.7 (±0.2)	0.7 (±0.2)	0.978

BMI, body mass index; TB, total bilirubin; INR, international normalized ratio; BIA, bioelectrical impedance analysis.

**Table 2 jpm-12-01755-t002:** Operative outcomes of patients.

	BIA Group (n = 21)	Control Group (n = 30)	*p*
Operative duration, min	240.5 (±45.6)	276.5 (±54.0)	0.016
Hepatectomy time, min	167.5 (±32.5)	203.9 (±42.6)	0.002
Intraoperative fluids			
Crystalloid, mL	2623 (±646)	2452 (±586)	0.329
Colloid, mL	183 (±242)	267 (±254)	0.245
Urine output, mL	306 (±170)	324 (±172)	0.707
Estimated blood loss, mL	325 (±212)	382 (±210)	0.349
Blood transfusion	0	0	-
Inotropics usage	9 (39.1%)	9 (32.1%)	0.603
Pleural effusion	2 (9.5%)	1 (3.3%)	0.355
AKI	1 (4.8%)	2 (7.1%)	0.731
Postoperative liver insufficiency	0	2 (7.1%)	0.191
Major postoperative complication	0	0	-
Postoperative hospital stay, days	10.4 (±2.2)	10.8 (±3.6)	0.679

AKI, acute kidney injury; BIA, bioelectrical impedance analysis.

**Table 3 jpm-12-01755-t003:** Analysis of risk factors for cleaner surgical field.

	Univariate Analysis			Multivariate Analysis		
Variable	OR	95% CI	*p*	OR	95% CI	*p*
Age (yrs)	0.994	0.945–1.045	0.809			
Male	0.711	0.228–2.220	0.557			
Diabetes mellitus	1.227	0.073–20.763	0.887			
Hypertension			-			
BMI ≥ 25	2.308	0.703–7.570	0.168			
Macrosteatosis ≥ 10%			-			
Remnant volume < 30%	2.571	0.218–30.318	0.453			
Baseline liver function						
TB (mg/dL)	0.232	0.014–3.910	0.311			
Albumin (g/dL)	0.671	0.118–4.041	0.681			
INR			-			
Preoperative BIA	4.318	1.296–14.383	0.017	6.914	1.699–28.191	0.012

BMI, body mass index; TB, Total bilirubin; INR, international normalized ratio; BIA, bioelectrical impedance analysis; OR, odds ratio; CI, confidence interval.

## Data Availability

Data available from the authors upon request.

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
