# Peer review of "Bioelectrical Impedance Analysis for Preoperative Volemia Assessment in Living Donor Hepatectomy"

_jpm, 2022, doi:10.3390/jpm12111755_

Round 1

Reviewer 1 Report

Comments for the author

Suh SW performed an interesting study accessing the usefulness of bioelectrical impedance assessment in living donor hepatectomy. The manuscript is well-written, however some issues must be addressed.

1)      In the abstract, the sentence “Preoperative BIA was the only significant predictor for a cleaner surgical field (odds ratio, 6.914; 95% confidence interval, 1.6985–28.191, P = 0.007), significantly associated with decreased operative duration (P = 0.001) and estimated blood loss (P < 0.001)” is misleading, since preoperative BIA was not associated with decreased blood loss. It was the cleaner operation field actually. Please rewrite this sentence to make it clearer.

2)      There is a similar sentence in the conclusion (292-293), which should be rewritten as well.

3)      I suggest the author to mention in the abstract that ECT/TBW < 0.378 was correlated to CVP < 5mmHg in a previous study.

4)      What does “Early NPO” mean in Figure 1? This term was not used elsewhere in the text. I suggest authors to use whole-day NPO, as it was in the text

5)      Please use the name of statistical tests instead of mathematical terms.

6)      Please state the meaning of SPSS abbreviation

7)      I suggest authors to remove figure 2, since the text in the results section provides the same information

8)      What does NS mean in table 3?

9)      The size of figure 5 could be reduced and all subfigures (A, B, C) could be merged together to maximize space utilization.

10)   In the discussion, the author mentions “differences in the study population” and comments about HCC patients (245-248). However, there were no differences between groups in table 1 and there is no cases of HCC, as all patients were living donors. Please clarify these issues.

11)   In the discussion, sentence 261-262 is confusing. Please rewrite it.

Author Response

Suh SW performed an interesting study accessing the usefulness of bioelectrical impedance assessment in living donor hepatectomy. The manuscript is well-written, however some issues must be addressed.

Ans) I’m thank the reviewer for going through our manuscript carefully and suggesting points to improve the same.

  • In the abstract, the sentence “Preoperative BIA was the only significant predictor for a cleaner surgical field (odds ratio, 6.914; 95% confidence interval, 1.6985–28.191, P = 0.007), significantly associated with decreased operative duration (P = 0.001) and estimated blood loss (P < 0.001)” is misleading, since preoperative BIA was not associated with decreased blood loss. It was the cleaner operation field actually. Please rewrite this sentence to make it clearer.

Ans) I rewrite this sentence in the abstract to make it clearer as below.

‘A cleaner surgical field was significantly associated with decreased operative duration (P = 0.001) and estimated blood loss. Preoperative BIA was the only significant predictor a cleaner surgical field (odds ratio, 6.914; 95% confidence interval, 1.6985–28.191, P = 0.007).’ (20-23)

  • There is a similar sentence in the conclusion (292-293), which should be rewritten as well.

Ans) I also rewrite this sentence in the discussion section.

‘A cleaner surgical field was significantly associated with decreased operative duration and estimated blood loss. Preoperative BIA was the only significant predictor a cleaner surgical field.’ (285-287)

  • I suggest the author to mention in the abstract that ECT/TBW < 0.378 was correlated to CVP < 5mmHg in a previous study.

Ans) I added the comment of correlation between ECT/TBW < 0.378 and CVP < 5mmHg in the abstract as below.

‘ECT/TBW < 0.378 was correlated to CVP < 5mmHg in a previous study that we used this value for personalized preoperative management.’ (14-15)

  • What does “Early NPO” mean in Figure 1? This term was not used elsewhere in the text. I suggest authors to use whole-day NPO, as it was in the text

Ans) I changed “Early NPO” to ‘Whole-day NPO in Figure 1.

  • Please use the name of statistical tests instead of mathematical terms.

Ans) I revised the statistical analysis of method section as below.

‘Clinico-demographic characteristics of living donors, changes of CVPs during opera-tion and intraoperative and perioperative outcomes of patients were compared using Stu-dent’s t-test for distributed data, presented as means ± standard deviations and χ2 test for descriptive data. The surgical field grades were compared using the Mann-Whitney U test. Univariate and multivariate analysis of risk factors for cleaner surgical field was per-formed using an ordinary logistic regression model. Differences in operative outcomes according to the surgical field grade were compared using Student’s t-test. P values < 0.05 were considered to indicate statistical significance. Statistical analysis was conducted us-ing statistical package for the social sciences (SPSS) version 19.0 (IBM Corp., Armonk, NY, USA).’ (137-146)

  • Please state the meaning of SPSS abbreviation

Ans) I state the SPSS abbreviation in the method section as below.

‘statistical package for the social sciences (SPSS)’ (145)

  • I suggest authors to remove figure 2, since the text in the results section provides the same information

Ans) I removed figure 2.

  • What does NS mean in table 3?

Ans) NS means not significant, but it might be wrong. Because I used this term when the statistical analysis was not possible due to small sample size. Instead of NS, blank or – maybe better to prevent misunderstanding. I changed all of ‘NS’ to ‘-‘ in Table 2 and 3.

  • The size of figure 5 could be reduced and all subfigures (A, B, C) could be merged together to maximize space utilization.

Ans) I changed figure 5 considering reviewer’s opinion.

  • In the discussion, the author mentions “differences in the study population” and comments about HCC patients (245-248). However, there were no differences between groups in table 1 and there is no cases of HCC, as all patients were living donors. Please clarify these issues.

Ans) It means differences in the study population of previous studies with HCC patients, but this sentence might be confusing. I revised the comments as below.

‘Finally, the living donors had a normal healthy liver which make hepatic parenchymal dissection and bleeding control more easy than other patients.’ (241-243)

11)   In the discussion, sentence 261-262 is confusing. Please rewrite it.

Ans) This sentence is confusing and seems to be not necessary. I deleted this sentence.

Reviewer 2 Report

In this study, Suk-Won Suh reported using a non-invasive test, bioelectrical impedance analysis (BIA), in the preoperative assessment in living donor hepatectomy. In this study, a ratio of extracellular water: total body water (ECW/TBW) was measured and a cut-point value, 0.378, was applied to evaluate preoperative volume. An improvement of surgical outcome was found in predicting central venous pressure, decreased operation time and a better surgical field grade in living donor hepatectomy. This is an interesting and practical study in the clinical application. I have some comments and questions as below:

1.      Because this is a retrospective study, patients in control group (N=30) received the operation before the patients in BIA group (N=21). By increasing the experience in the operation of living donor hepatectomy, just like a learning curve, the operation time and surgical field scale may be improved without the help of BIA test.

2.      Could author give the data about the correlation between preoperative ECW/TBW value and CVP value in this study?

3.      Table 1 reveals the mean values of hepatic steatosis and macrosteatosis>=10%. How are these data collected? Are these data correlated with any other BIA data, such as body fat component or phase angle?

4.      There was no DM patient in the BIA group (Table 1). According to study protocol, donors with ECW/TBW >= 0.378 would receive whole-day NPO. How to prevent hypoglycemia event if a DM donor with ECW/TBW >=0.378? Set IV fluid with glucose water?

5.      Table 2 shows the mean operation duration time (minutes). Could the operation time be separated into captured donor liver time and total time? The operation time of living donor right hepatectomy may be lower in patients with low CVP or ECW/TBW <0.378.

6.      There were two donors in the BIA group whose ECW/TBW values kept more than 0.378 even after whole day NPO. Was their operation time more than other cases in the BIA group whose ECW/TBW less than 0.378 after whole day NPO? 

Author Response

In this study, Suk-Won Suh reported using a non-invasive test, bioelectrical impedance analysis (BIA), in the preoperative assessment in living donor hepatectomy. In this study, a ratio of extracellular water: total body water (ECW/TBW) was measured and a cut-point value, 0.378, was applied to evaluate preoperative volume. An improvement of surgical outcome was found in predicting central venous pressure, decreased operation time and a better surgical field grade in living donor hepatectomy. This is an interesting and practical study in the clinical application. I have some comments and questions as below:

Ans) I’m thank the reviewer for going through our manuscript carefully and suggesting points to improve the same.

  1. Because this is a retrospective study, patients in control group (N=30) received the operation before the patients in BIA group (N=21). By increasing the experience in the operation of living donor hepatectomy, just like a learning curve, the operation time and surgical field scale may be improved without the help of BIA test.

Ans) I agree with your opinion that operative outcomes might be also influenced by the surgeon’s experience. It might be a limitation of this study and further large-scale prospective studies are required to confirm the usefulness of BIA in living donor hepatectomy. We added these comments as a limitation in the discussion section.

‘Living donors in control group received the operation before those in BIA group that the experience in the operation might have influenced on operative outcomes.’ (276-278)

  1. Could author give the data about the correlation between preoperative ECW/TBW value and CVP value in this study?

Ans) In this study, 12 of 21 donors had repeated BIA on the operative day that only these results could be correlated with CVP value, evaluated on the same day. Because of small sample size and narrow range of CVP values of these donors, 3 to 7 mmHg, correlation between preoperative ECW/TBW value and CVP value cannot be statistically analyzed. Further large-scale studies would be required.

  1. Table 1 reveals the mean values of hepatic steatosis and macrosteatosis>=10%. How are these data collected? Are these data correlated with any other BIA data, such as body fat component or phase angle?

Ans) We retrospectively collected this data using pathologic reports of donors.

  1. There was no DM patient in the BIA group (Table 1). According to study protocol, donors with ECW/TBW >= 0.378 would receive whole-day NPO. How to prevent hypoglycemia event if a DM donor with ECW/TBW >=0.378? Set IV fluid with glucose water?

Ans) It is a difficult question. DM patient needs glucose absorption, but surgeon want to restrict fluid. I think small volume of high concentration glucose could be considered to satisfy both requirements.

  1. Table 2 shows the mean operation duration time (minutes). Could the operation time be separated into captured donor liver time and total time? The operation time of living donor right hepatectomy may be lower in patients with low CVP or ECW/TBW <0.378.

Ans) We re-calculated the captured graft time using operative data. There was also significant difference in captured graft time between the groups. I added this data in the results section and Table 2.

‘The mean operative duration (240.5 ± 45.6 vs. 276.5 ± 54.0 min, p = 0.016) and hepa-tectomy time were significantly shorter in the BIA group than in the control group (2167.5 ± 32.5 vs. 203.9 ± 42.6 min, p = 0.002).’ (177-179)

  1. There were two donors in the BIA group whose ECW/TBW values kept more than 0.378 even after whole day NPO. Was their operation time more than other cases in the BIA group whose ECW/TBW less than 0.378 after whole day NPO?

Ans) There was no significant difference of operative duration between two donors with a ECW/TBW values ≥ 0.378 even after whole day NPO and others with a ECW/TBW values < 0.378 (232.5 ± 38.9 vs. 233.5 ± 45.5 min, p = 0.978). CVP values of two donors were 6~7mmHg and others have 3~5 mmHg that this small difference might have not make significant difference in statistical analysis
